# DPO-Diff: On Discrete Prompt Optimization for Text-to-Image Diffusion Models

## Abstract

This paper introduces the first gradient-based framework for prompt optimization in text-to-image diffusion models. We formulate prompt engineering as a discrete optimization problem over the language space. Two major challenges arise in efficiently finding a solution to this problem: *1) Enormous Domain Space:* Setting the domain to the entire language space poses significant difficulty to the optimization process. *2) Text Gradient:* Computing the text gradient incurs prohibitively high memory-runtime complexity, as it requires backpropagating through all inference steps of the diffusion model. Beyond the problem formulation, our main technical contributions lie in solving the above challenges. First, we design a family of dynamically generated compact subspaces comprised of only the most relevant words to user input, substantially restricting the domain space. Second, we introduce "Shortcut Gradient" — an effective replacement for the text gradient that can be obtained with constant memory and runtime. Empirical evaluation on prompts collected from diverse sources (DiffusionDB, ChatGPT, COCO) suggests that our method can discover prompts that substantially improve (prompt enhancement) or destroy (adversarial attack) the faithfulness of images generated by the text-to-image diffusion model.

## 1 Introduction

Large-scale text-based generative models exhibit a remarkable ability to generate novel content conditioned on user input prompts (Ouyang et al., 2022; Touvron et al., 2023; Rombach et al., 2022; Ramesh et al., 2022; Saharia et al., 2022; Ho et al., 2022; Yu et al., 2022; Chang et al., 2023). Despite being trained with huge corpora, there still exists a substantial gap between user intention and what the model interprets (Zhou et al., 2022; Feng et al., 2022; Rombach et al., 2022; Radford et al., 2021; Lian et al., 2023; Ouyang et al., 2022; Ramesh et al., 2022). The misalignment is even more severe in text-to-image generative models, partially since they often rely on much smaller and less capable text encoders (Radford et al., 2021; Cherti et al., 2023; Raffel et al., 2020) than large language models (LLMs). As a result, instructing a large model to produce intended content often requires laborious human efforts in crafting the prompt through trials and errors (a.k.a. Prompt Engineering) (Art, Year; Wang et al., 2022; Witteveen & Andrews, 2022; Liu & Chilton, 2022; Zhou et al., 2022; Hao et al., 2022). To automate this process for language generation, several recent attempts have shown tremendous potential in utilizing LLMs to enhance prompts (Pryzant et al., 2023; Zhou et al., 2022; Chen et al., 2023; Guo et al., 2023; Yang et al., 2023; Hao et al., 2022). However, efforts on text-to-image generative models remain scarce and preliminary, probably due to the challenges faced by these models' relatively small text encoders in understanding subtle language cues.

**DPO-Diff.** This paper presents a systematic study of prompt optimization for text-to-image diffusion models. We introduce a novel optimization framework based on the following key observations. *1) Prompt engineering can be formulated as a Discrete Prompt Optimization (DPO) problem over the space of natural languages.* Moreover, the framework can be used to find prompts that either improve (prompt enhancement) or destroy (adversarial attack) the generation process, by simply reversing the sign of the objective function. *2) We show that for diffusion models with classifier-free guidance (Ho & Salimans, 2022), improving the image generation process is more effective when optimizing "negative prompts" (Andrew, 2023; Woolf, 2022) than positive prompts.* Beyond the problem formulation of DPO-Diff, where "Diff" highlights our focus on text-to-image diffusion

models, the main technical contributions of this paper lie in efficient methods for solving this optimization problem, including the design of compact domain spaces and a gradient-based algorithm.

**Compact domain spaces.** DPO-Diff's domain space is a discrete search space at the word level to represent prompts. While this space is generic enough to cover any sentence, it is excessively large due to the dominance of words irrelevant to the user input. To alleviate this issue, we design a family of dynamically generated compact search spaces based on relevant word substitutions, for both positive and negative prompts. These subspaces enable efficient search for both prompt enhancement and adversarial attack tasks.

**Shortcut gradients for DPO-Diff.** Solving DPO-Diff with a gradient-based algorithm requires computing the text gradient, i.e., backpropagating from the generated image, through all inference steps of a diffusion model, and finally to the discrete text. Two challenges arise in obtaining this gradient: 1) This process incurs compound memory-runtime complexity over the number of backward passes through the denoising step, making it prohibitive to run on large-scale diffusion models (e.g., a 870M-parameter Stable Diffusion v1 requires ∼750G memory to run backpropagation through 50 inference steps (Rombach et al., 2022)). 2) The embedding lookup tables in text encoders are non-differentiable. To reduce the computational cost in 1), we provide the first generic replacement for the text gradient that bypasses the need to unroll the inference steps in a backward pass, allowing it to be computed with constant memory and runtime. To backpropagate through the discrete embedding lookup table, we continuously relax the categorical word choices to a learnable smooth distribution over the vocabulary, using the Gumbel Softmax trick (Guo et al., 2021; Jang et al., 2016; Dong & Yang, 2019). The gradient obtained by this method, termed **Shortcut Gradient**, enables us to efficiently solve DPO-Diff regardless of the number of inference steps of a diffusion model.

To evaluate our prompt optimization method for the diffusion model, we collect and filter a set of challenging prompts from diverse sources including DiffusionDB (Wang et al., 2022), COCO (Lin et al., 2014), and ChatGPT (Ouyang et al., 2022). Empirical results suggest that DPO-Diff can effectively discover prompts that improve (or destroy for adversarial attack) the faithfulness of text-to-image diffusion models, surpassing human-engineered prompts and prior baselines by a large margin. We summarize our primary contributions as follows:

- **DPO-Diff:** A generic framework for prompt optimization as a discrete optimization problem over the space of natural languages, of arbitrary metrics.
- **Compact domain spaces:** A family of dynamic compact search spaces, over which a gradient-based algorithm enables efficient solution finding for the prompt optimization problem.
- **Shortcut gradients:** The first novel computation method to enable backpropagation through the diffusion models' lengthy sampling steps with constant memory-runtime complexity, enabling gradient-based search algorithms.
- **Negative prompt optimization:** The first empirical result demonstrating the effectiveness of optimizing negative prompts for diffusion models.

## 2 RELATED WORK

**Text-to-image diffusion models.** Diffusion models trained on a large corpus of image-text datasets significantly advanced the state of text-guided image generation (Rombach et al., 2022; Ramesh et al., 2022; Saharia et al., 2022; Chang et al., 2023; Yu et al., 2022). Despite the success, these models can sometimes generate images with poor quality. While some preliminary observations suggest that negative prompts can be used to improve image quality (Andrew, 2023; Woolf, 2022), there exists no principled way to find negative prompts. Moreover, several studies have shown that large-scale text-to-image diffusion models face significant challenges in understanding language cues in user input during image generation; Particularly, diffusion models often generate images with missing objects and incorrectly bounded attribute-object pairs, resulting in poor "faithfulness" or "relevance" (Hao et al., 2022; Feng et al., 2022; Lian et al., 2023; Liu et al., 2022). Existing solutions to this problem include compositional generation (Liu et al., 2022), augmenting diffusion model with large language models (Yang et al., 2023), and manipulating attention masks (Feng et al., 2022). As a method orthogonal to them, our work reveals that negative prompt optimization can also alleviate this issue.

**Prompt optimization for text-based generative models.** Aligning a pretrained large language model (LLM) with human intentions is a crucial step toward unlocking the potential of large-scale text-based generative models (Ouyang et al., 2022; Rombach et al., 2022). An effective line of training-free alignment methods is prompt optimization (PO) (Zhou et al., 2022). PO originated from in-context learning (Dale, 2021), which is mainly concerned with various arrangements of task demonstrations. It later evolves into automatic prompt engineering, where powerful language models are utilized to refine prompts for certain tasks (Zhou et al., 2022; Pryzant et al., 2023; Yang et al., 2023; Pryzant et al., 2023; Hao et al., 2022). While PO has been widely explored for LLMs, efforts on diffusion models remain scarce. The most relevant prior work to ours is Promptist (Hao et al., 2022), which finetunes an LLM via reinforcement learning from human feedback (Ouyang et al., 2022) to augment user prompts with artistic modifiers (e.g., high-resolution, 4K) (Art, Year), resulting in aesthetically pleasing images. However, the lack of paired contextual-aware data significantly limits its ability to follow the user intention (Figure 2b).

**Backpropagating through the sampling steps of diffusion models.** Text-to-image diffusion models generate images via a progressive denoising process, making multiple passes through the same network (Ho et al., 2020). When a loss is applied to the output image, computing the gradient w.r.t. any model component (text, weight, sampler, etc.) requires backpropagating through all the sampling steps. This process incurs compound complexity over the number of backward passes in both memory and runtime, making it infeasible to run on regular commercial devices. Existing efforts achieve constant memory via gradient checkpointing (Watson et al., 2021) or solving an augmented SDE problem (Nie et al., 2022), at the expense of even higher runtime. In this paper, we propose a novel solution to compute a "shortcut" gradient, resulting in constant complexity in both memory and runtime.

## 3 PRELIMINARIES ON DIFFUSION MODEL

We provide a brief overview of relevant concepts in diffusion models, and refer the reader to (Luo, 2022) for detailed derivations.

**Denoising diffusion probablistic models.** On a high level, diffusion models (Ho et al., 2020) are a type of hierarchical variational autoencoder (Sønderby et al., 2016) that generates samples by reversing a progressive noising process. Let $\boldsymbol{x}_0 \cdots \boldsymbol{x}_T$ be a series of intermediate samples at increasing noise levels, the noising (forward) process can be expressed as the following Markov chain:

$$q(\boldsymbol{x}_t|\boldsymbol{x}_{t-1}) = \mathcal{N}(\boldsymbol{x}_t; \sqrt{1-\beta_t}\boldsymbol{x}_{t-1}, \beta_t\boldsymbol{I}) \quad t = 1 \sim T, \tag{1}$$

where $\beta$ is a scheduling variable. Using Gaussian reparameterization, sampling $\boldsymbol{x}_t$ from $\boldsymbol{x}_0$ can be completed in a single step:

$$\boldsymbol{x}_t = \sqrt{\bar{\alpha}_t}\boldsymbol{x}_0 + \sqrt{1-\bar{\alpha}_t}\epsilon, \quad \alpha_t = 1 - \beta_t \text{ and } \bar{\alpha}_t = \prod_{i=1}^{t}\alpha_i, \tag{2}$$

where $\epsilon$ is a standard Gaussian error. The reverse process starts with a standard Gaussian noise, $\boldsymbol{x}_T \sim \mathcal{N}(\boldsymbol{0}, \boldsymbol{I})$, and progressively denoises it using the following joint distribution:

$$p_\theta(\boldsymbol{x}_{0:T}) = p(\boldsymbol{x}_T)\prod_{t=1}^{T} p_\theta(\boldsymbol{x}_{t-1}|\boldsymbol{x}_t) \text{ where } p_\theta(\boldsymbol{x}_{t-1}|\boldsymbol{x}_t) = \mathcal{N}(\boldsymbol{x}_{t-1}; \mu_\theta(\boldsymbol{x}_t, t), \boldsymbol{\Sigma}).$$

While the mean function $\mu_\theta(\boldsymbol{x}_t, t)$ can be parameterized by a neural network (e.g., UNet (Rombach et al., 2022; Ronneberger et al., 2015)) directly, prior studies found that modeling the residual error $\epsilon(\boldsymbol{x}_t, t)$ instead works better empirically Ho et al. (2020). The two strategies are mathematically equivalent as $\mu_\theta(\boldsymbol{x}_t, t) = \frac{1}{\sqrt{\alpha_t}}(\boldsymbol{x}_t - \frac{1-\alpha_t}{\sqrt{1-\bar{\alpha}_t}}\epsilon(\boldsymbol{x}_t, t))$.

**Classifier-free guidance for conditional generation.** The above formulation can be easily extended to conditional generation via classifier-free guidance (Ho & Salimans, 2022), widely adopted in contemporary diffusion models. At each sampling step, the predicted error $\tilde{\epsilon}$ is obtained by subtracting the unconditional error from the conditional error (up to a scaling factor $w$):

$$\tilde{\epsilon}_\theta(\boldsymbol{x}_t, c(s), t) = (1+w)\epsilon_\theta(\boldsymbol{x}_t, c(s), t) - w\epsilon_\theta(x_t, c(""), t), \tag{3}$$

where $c(s)$ is the conditional signal of text $s$, and the unconditional prior $c(\text{""})$ is obtained by passing an empty string to the text encoder. If we replace this empty string with an actual text, then it becomes a "negative prompt" (Andrew, 2023; Woolf, 2022), indicating what to exclude from the generated image.

# 4 DPO-DIFF: DISCRETE PROMPT OPTIMIZATION FOR DIFFUSION MODELS

This section lays out the components of DPO-Diff framework. Section 4.1 explains how to formulate the problem into optimization over the text space. This is followed by the full algorithm for solving this optimization, including the compact search space in Section 4.2, and the gradient-based search algorithm in Section 4.3.

## 4.1 FRAMEWORK OVERVIEW

Our main insight is that prompt engineering can be formulated as a discrete optimization problem in the language space, called DPO-Diff. Concretely, we represent the problem domain $\mathcal{S}$ as a sequence of $M$ words $w_i$ from a predefined vocabulary $\mathcal{V}$: $\mathcal{S} = \{w_1, w_2, \ldots w_M | \forall i, \; w_i \in \mathcal{V}\}$. This space is generic enough to cover all possible sentences of lengths less than $M$ (when the empty string is present). Let $G(s)$ denote a text-to-image generative model, and $s_{user}$, $s$ denote the user input and optimized prompt, respectively. The optimization problem can be written as

$$\min_{s \in \mathcal{S}} \mathcal{L}(G(s), s_{user}) \qquad \text{s.t.} \quad d(s, s_{user}) \leq \lambda, \tag{4}$$

where $\mathcal{L}$ can be any objective function that measures the effectiveness of the learned prompt when used to generate images, and $d(\cdot, \cdot)$ is an optional constraint function that restricts the distance between the optimized prompt and the user input. Following previous works (Hao et al., 2022), we use clip loss $\text{CLIP}(I, s_{user})$ ("crumb", 2022) to measure the alignment between the generated image $I$ and the user prompt $s_{user}$ — the **Faithfulness** of the generation process. Like any automatic evaluator for generative models, the clip score is certainly not free from errors. However, through the lens of human evaluation, we find that it is mostly aligned with human judgment for our task.

This DPO-Diff framework is versatile for handling both prompt improvement and adversarial attack. Finding adversarial prompts can help diagnose the failure modes of generative models, as well as augment the training set to improve a model's robustness via adversarial training (Madry et al., 2017). We define adversarial prompts for text-to-image generative models as follows.

**Definition 4.1.** Given a user input $s_{user}$, an adversarial prompt $s_{adv}$ is a text input that is semantically similar to $s_{user}$, yet causes the model to generate images that cannot be described by $s_{user}$.

Intuitively, Definition refadv aims at perturbing the user prompt without changing its overall meaning to destroy the prompt-following ability of image generation. Formally, the adversarial prompt is a solution to the following problem,

$$\min_{s \in \mathcal{S}} -\mathcal{L}(G(s), s_{user}) \quad \text{s.t.} \; d(s, s_{user}) \leq \lambda \tag{5}$$

where the first constraint enforces semantic similarity.

To apply DPO-Diff to adversarial attack, we can simply add a negative sign to $\mathcal{L}$, and restrict the distance between $s$ and $s_{user}$ through $d$. This allows equation 5 to produce an $s$ that increases the distance between the image and the user prompt, while still being semantically similar to $s_{user}$.

## 4.2 COMPACT SEARCH SPACE DESIGN FOR EFFICIENT PROMPT DISCOVERY

While the entire language space facilitates maximal generality, it is also unnecessarily inefficient as it is popularized with words irrelevant to the task. We propose a family of compact search spaces that dynamically extracts a subset of task-relevant words to the user input.

### 4.2.1 APPLICATION 1: DISCOVERING ADVERSARIAL PROMPTS FOR MODEL DIAGNOSIS

**Synonym Space.** In light of the first constraint on semantic similarity in equation 5, we build a search space for the adversarial prompts by substituting each word in the user input $s_{user}$ with its synonyms (Alzantot et al., 2018), preserving the meaning of the original sentence. The synonyms can be found by either dictionary lookup or querying ChatGPT (see Appendix C.2 for more details).

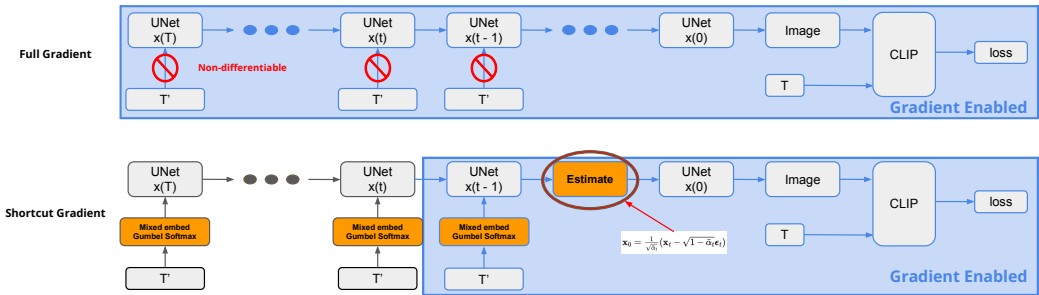

Figure 1: Computational procedure of Shortcut Gradient (Bottom) v.s. Full Gradient (Top) on text.

### 4.2.2 APPLICATION 2: DISCOVERING ENHANCED PROMPTS FOR IMAGE GENERATION

While the Synonym Space is suitable for attacking diffusion models, we found that it performs poorly on finding improved prompts. This is in contradiction to LLMs where rephrasing user prompts can often lead to substantial gains (Zhou et al., 2022). One plausible reason is that contemporary diffusion models often rely on a small-scale clip-based text encoders (Radford et al., 2021; Cherti et al., 2023; Raffel et al., 2020), which are weaker than LLMs with many known limitations in understanding subtle language cues (Feng et al., 2022; Liu et al., 2022; Yang et al., 2023).

Inspired by these observations, we propose a novel solution to optimize for **negative** prompts instead — a unique concept that rises from classifier-free guidance (Ho & Salimans, 2022) used in diffusion models (Section 3). To the best of our knowledge, we provide the first exploratory work on automated negative prompt optimization.

**Antonym Space.** We propose to build the space of negative prompts based on the antonyms of each word, as opposed to the Synonym Space for adversarially attacking the model. Intuitively, the model's output image can safely exclude the content with the opposite meaning to the words in the user input, so it instead amplifies the concepts presented in the positive prompt. Similar to synonyms, the antonyms of words in a user prompt can be obtained via dictionary lookup or ChatGPT.

**Negative Prompt Library (NPLib).** We further crawl and filter a library of human-crafted generic negative prompts to augment the antonym space. This augmentation enhances the image generation quality and provides a safeguard when a user input has a small number of high-quality antonyms. We term our library **NPLib**, which will be released with our codebase.

### 4.3 A GRADIENT-BASED ALGORITHM AS A DPO-DIFF SOLVER

Due to the query efficiency of white-box algorithms leveraging gradient information, we also explore a gradient-based method to solve equation 4 and equation 5. However, obtaining this text gradient is non-trivial due to two major challenges. 1) Backpropagating through the sampling steps of the diffusion inference process incurs high complexity w.r.t. memory and runtime, making it prohibitively expensive to obtain gradients. [1] 2) The embedding lookup table used in the text encoder is non-differentiable. Section 4.3.1 introduces the Shortcut Gradient, a replacement for text gradient with constant memory and runtime. Section 4.3.2 discusses how to backpropagate through the embedding lookup table via continuous relaxation. Section 4.3.3 describes how to sample from the learned distribution via evolutionary search.

### 4.3.1 BACKPROPAGATING THROUGH DIFFUSION SAMPLING STEPS

**Shortcut gradient.** Backpropagating through the diffusion model inference process requires executing multiple backward passes through the generator network (Watson et al., 2021; Nie et al., 2022); For samplers with 50 inference steps (e.g., DDIM (Song et al., 2020)), it raises the runtime and memory cost by **50 times** compared to a single diffusion training step. To alleviate this issue, we propose Shortcut Gradient, an efficient replacement for text gradients that can be obtained with constant memory and runtime.

---

[1]The text gradient is completely different from "Textual Inversion" (Gal et al., 2022), as the later can be obtained in the same way as regular gradients used in diffusion training, requiring only a single backward pass.

The key idea behind the Shortcut Gradient is to reduce gradient computation from all to $K$ sampling steps, resulting in a constant number of backward passes. The entire pipeline (Figure 1) can be divided into three steps:

*(1) Sampling without gradient from step $T$ (noise) to $t$.* In the first step, we simply disable gradients up to step $t$. No backward pass is required for this step.

*(2) Enable gradient from $t$ to $t - K$.* We enable the computational graph for a backward pass for $K$ step starting at $t$.

*(3) Estimating $\boldsymbol{x}_0$ from $\boldsymbol{x}_{t-K}$ through closed-form solution.* To bypass the gradient computation in the remaining steps, simply disabling the gradient like (1) is no longer valid because otherwise the loss applied to $\boldsymbol{x}_0$ could not propagate back. Directly decoding and feeding the noisy intermediate image $\boldsymbol{x}_{t-K}$ to the loss function is also not optimal due to distribution shift (Dhariwal & Nichol, 2021). Instead, we propose to use the current estimate of $\boldsymbol{x}_0$ from $\boldsymbol{x}_{t-K}$ to bridge the gap. From the forward equation of the diffusion model, we can derive a connection between the final image $\hat{\boldsymbol{x}}_0$ and $\boldsymbol{x}_{t-K}$ as $\hat{\boldsymbol{x}}_0 = \frac{1}{\sqrt{\bar{\alpha}_{t-K}}}(\boldsymbol{x}_{t-K} - \sqrt{1 - \bar{\alpha}_{t-K}}\boldsymbol{\epsilon}_{t-K})$. In this way, the Jacobian of $\hat{\boldsymbol{x}}_0$ w.r.t. $\boldsymbol{x}_{t-K}$ can be computed analytically, with complexity independent of $t$.

- **Remark 1**: The estimation is not a trick — it directly comes from a mathematically equivalent interpretation of the diffusion model, where each inference step can be viewed as computing $\hat{\boldsymbol{x}}_0$ and plugging it into $q(\boldsymbol{x}_{t-K}|\boldsymbol{x}_t, \hat{\boldsymbol{x}}_0)$ to obtain the transitional probability.

- **Remark 2**: The computational cost of the Shortcut gradient is controlled by $K$. Moreover, when we set $t = T$ and $K = T$, it becomes the full-text gradient.

**Strategy for selecting $t$.** At each iteration, we select a $t$ and compute the gradient of the loss over text embeddings using the above mechanism. Empirically, we found that setting it around the middle point and progressively reducing it produces the most salient gradient signals (Appendix C.1).

### 4.3.2 BACKPROPAGATING THROUGH EMBEDDINGS LOOKUP TABLE

In diffusion models, a tokenizer transforms text input into indices, which will be used to query a lookup table for corresponding word embeddings. To allow further propagating gradients through this non-differentiable indexing operation, we relax the categorical choice of words into a continuous probability of words. It can be viewed as learning a "distribution" over words. We parameterize the distribution using Gumbel Softmax (Jang et al., 2016) with uniform temperature ($\eta = 1$):

$$\tilde{e} = \sum_{i=1}^{|\mathcal{V}|} p(w = i; \alpha)e_i \ , \ \ p(w = i; \alpha) = \frac{\exp\left((\log \alpha_i + g_i)/\eta\right)}{\sum_{i=1}^{|\mathcal{V}|} \exp\left((\log \alpha_i + g_i)/\eta\right)}, \tag{6}$$

where $\alpha$ (a $|\mathcal{V}|$-dimensional vector) denotes the learnable parameter, $g$ denotes the Gumbel random variable, $e_i$ is the embedding of word $i$, and $\tilde{e}$ is the mixed embedding.

### 4.3.3 EFFICIENT SAMPLING WITH EVOLUTIONARY SEARCH

After learning a distribution over words, we can further sample candidate prompts from it via random search. However, random search is sample inefficient, as evidenced in AutoML literatures (Wu et al., 2019). We thus adopt an evolutionary algorithm (Goldberg, 1989) to search for the best candidate instead, which is simple to implement yet demonstrates strong performance. We view sentences as DNAs and the word choices as nucleotides; To initiate the evolution process, we fill the population with samples from the learned distribution and apply a traditional Evolution Algorithm to find the best one. The complete algorithm for **DPO-Diff** can be found in Algorithm 1 in Appendix B.

**Remark: Blackbox Optimization.** When the internal state of the model is accessible (e.g., the model owner provides prompt suggestions), gradient information can greatly speed up the search process. In cases where only forward API is available, our Evolutionary Search (ES) module can be used as a stand-alone black-box optimizer, thereby extending the applicability of our framework. We ablate this choice in Section 6.1, where ES archives descent results given enough queries.

Figure 2: Images generated by user input and improved negative prompts using Stable Diffusion. More examples can be found in Appendix E.

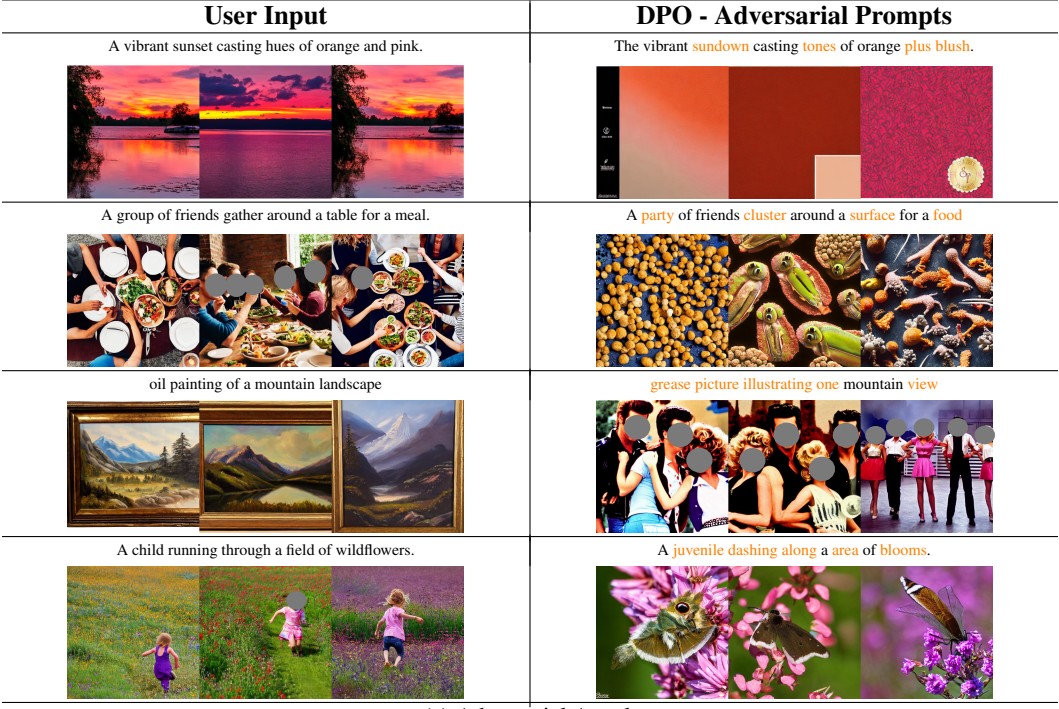

(a) Adversarial Attack

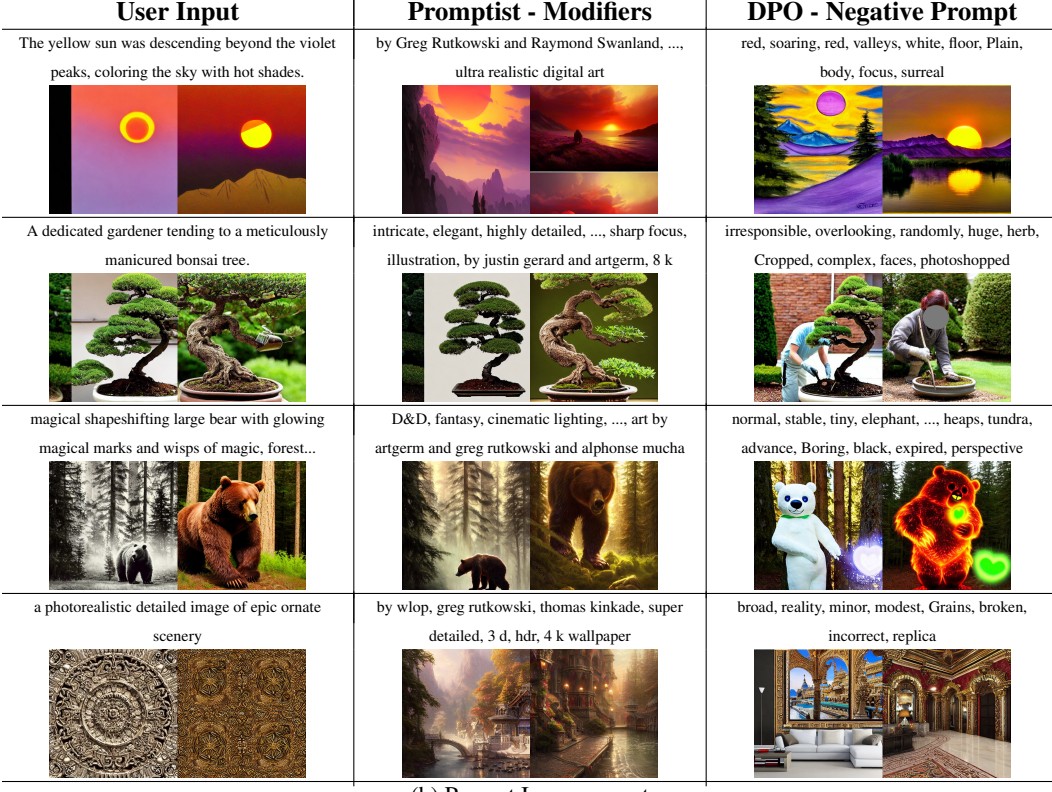

(b) Prompt Improvement

## 5 EXPERIMENTS

### 5.1 EXPERIMENTAL SETUP

**Dataset preparation.** To encourage semantic diversity, we collect a prompt dataset from three sources: DiffusionDB (Wang et al., 2022), ChatGPT generated prompts (Ouyang et al., 2022), and COCO (Lin et al., 2014). For each source, we filter 100 **"hard prompts"** with a clip loss higher (lower for adversarial attack) than a threshold, amounting to **600 prompts** in total for two tasks. Due to space limit, we include preparation details in Appendix D.1.

**Evaluation.** All methods are evaluated quantitatively using the clip loss (Crowson et al., 2022), complemented by qualitative evaluation by human judgers. We select Stable Diffusion v1-4 as the base model. Each prompt is evaluated under three random seeds (shared across different methods). The human evaluation protocol can be found in Appendix D.2.

**Optimization Parameters.** We use the spherical clip loss ("crumb", 2022) as the objective function, which ranges between 0.75 and 0.85 for most inputs. The $K$ for the shortcut gradient is set to 1 since we found that it already produces effective supervision signals. To generate the search spaces, we prompt ChatGPT for at most 5 substitutes of each word in the user prompt. Furthermore, we use a fixed set of hyperparameters for both prompt improvement and adversarial attacks. We include a detailed discussion on all the hyperparameters and search space generation in Appendix C.

### 5.2 DISCOVERING ADVERSARIAL PROMPTS

Unlike RLHF-based methods for enhancing prompts (e.g., Promptist (Hao et al., 2022)) that requires fine-tuning a prompt generator when adapting to a new task, DPO-Diff can be seamlessly applied to finding adversarial prompts by simply reversing the sign of the objective function. These adversarial prompts can be used to diagnose the failure modes of diffusion models or improve their robustness via adversarial training (Madry et al., 2017).

Table 1a shows adversarial prompt results. Our method is able to perturb the original prompt to adversarial directions, resulting in a substantial increase in the clip loss. Figure 2a (more in 5) visualizes a set of intriguing images generated by the adversarial prompts. We can see that DPO-Diff can effectively explore the text regions where Stable Diffusion fails to interpret.

Table 1: Quantitative evaluation of DPO discovered prompts. For each method, we report the average spherical clip loss of the generated image and user input over all prompts. Note that spherical clip loss normally ranges from 0.75 - 0.85, hence a change above 0.05 is already substantial.

| Prompt | DiffusionDB | COCO | ChatGPT |
|---|---|---|---|
| User Input | 0.76 ± 0.03 | 0.77 ± 0.03 | 0.77 ± 0.02 |
| DPO-Adv | **0.86 ± 0.05** | **0.94 ± 0.04** | **0.95 ± 0.05** |

(a) Adversarial Attack ↑

| Prompt | DiffusionDB | COCO | ChatGPT |
|---|---|---|---|
| User Input | 0.87 ± 0.02 | 0.87 ± 0.01 | 0.84 ± 0.01 |
| Manual | 0.89 ± 0.04 | 0.88 ± 0.02 | 0.86 ± 0.03 |
| Promptist | 0.88 ± 0.02 | 0.87 ± 0.03 | 0.85 ± 0.02 |
| DPO | **0.81 ± 0.03** | **0.82 ± 0.02** | **0.78 ± 0.03** |

(b) Prompt Improvement ↓

### 5.3 PROMPT OPTIMIZATION FOR IMPROVING STABLE DIFFUSION

In this section, we apply DPO-Diff to discover refined prompts to improve the relevance of generated images with user intention. We compare our method with three baselines: (1) User Input. (2) Human Engineered Prompts (available only on DiffusionDB) (Wang et al., 2022). (3) Promptist (Hao et al., 2022), trained to mimic the human-engineered prompt provided in DiffusionDB. For DiffusionDB, following Promptist (Hao et al., 2022), we extract user input by asking ChatGPT to remove all trailing aesthetics from the original human-engineered prompts.

Table 1b summarizes the result. We found that both human-engineered and Promptist-optimized prompts do not improve the relevance. The reason is that they change the user input by merely adding a set of aesthetic modifiers to the original prompt, which are irrelevant to the semantics of user input and cannot improve the generated images' faithfulness to user intentions. This can be further evidenced by the examples in Figure 2b.

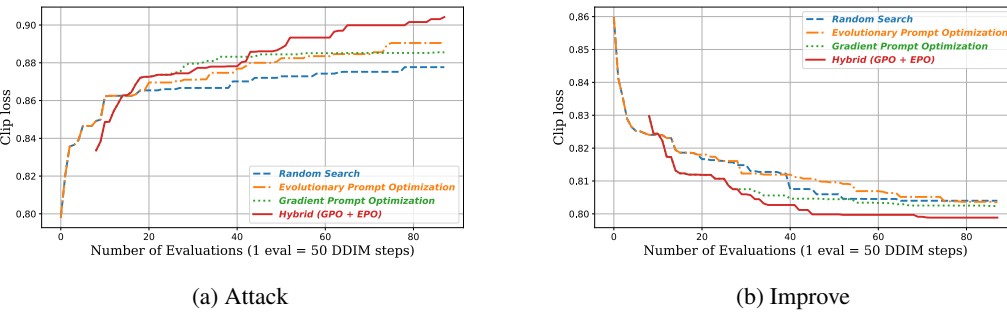

(a) Attack                                        (b) Improve

Figure 3: Learning curves of different search algorithms in solving DPO.

**Human Evaluation.**    We further asks 5 human judgers to evaluate the generated images of each method on a manually filtered subset of 100 prompts (see Appendix D.2 for the filtering and evaluation protocols). When evaluated based on how well the generated image can be described by the user input, the prompts discovered by DPO-Diff achieved a 64% win rate, 15% draw, and 21% loss rate compared with Promptist.

## 6    ABLATION STUDY

We conduct ablation studies on DPO-Diff using 30 randomly selected prompts (10 from each source). Each algorithm is run with 4 seeds to account for the randomness in the search phase.

### 6.1    COMPARISON OF DIFFERENT SEARCH ALGORITHMS.

We compare four search algorithms for DPO-Diff: Random Search (RS), Evolution Prompt Optimization (EPO), Gradient-based Prompt Optimization (GPO), and the full algorithm (GPO + ES). Figure 3 shows their performance under different search budgets (number of evaluations)[2]. While GPO tops EPO under low budgets, it also plateaus quicker as randomly drawing from the learned distribution is sample-inefficient. Combining GPO with EPO achieves the best overall performance.

### 6.2    NEGATIVE PROMPT V.S. POSITIVE PROMPT OPTIMIZATION

One finding in our work is that optimizing negative prompts (Antonyms Space) is more effective than positive prompts (Synonyms Space) for Stable Diffusion.   To verify the strength of these spaces, we randomly sample 100 prompts for each space and compute their average clip loss of generated images. Table 2 suggests that

Table 2: Quantitative evaluation of optimizing negative prompts (w/ Antonyms Space) and positive prompts (w/ Synonym Space) for Stable Diffusion.

| Prompt | DiffusionDB | ChatGPT | COCO |
|---|---|---|---|
| User Input | 0.8741 ± 0.0203 | 0.8159 ± 0.0100 | 0.8606 ± 0.0096 |
| Positive Prompt | 0.8747 ± 0.0189 | 0.8304 ± 0.0284 | 0.8624 ± 0.0141 |
| Negative Prompt | **0.8579 ± 0.0242** | **0.8133 ± 0.0197** | **0.8403 ± 0.0210** |

Antonyms Space contains candidates with consistently lower clip loss than Synonyms Space.

## 7    CONCLUSIONS

This work presents DPO-Diff, the first gradient-based framework for optimizing discrete prompts. We formulate prompt optimization as a discrete optimization problem over the text space. To improve the search efficiency, we introduce a family of compact search spaces based on relevant word substitutions, as well as design a generic computational method for efficiently backpropagating through the diffusion sampling process. DPO-Diff is generic - We demonstrate that it can be directly applied to effectively discover both refined prompts to aid image generation and adversarial prompts for model diagnosis. We hope that the proposed framework helps open up new possibilities in developing advanced prompt optimization methods for text-based image generation tasks. To motivate future work, we discuss the known limitations of DPO in Appendix A

---

[2]Since the runtime of backpropagation through one-step diffusion sampling is negligible w.r.t. the full sampling process (50 steps for DDIM sampler), we count it the same as one inference step.

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
