# OpenReview forum: "DPO-Diff: On Discrete Prompt Optimization of Text-to-Image Diffusion Models"
_ICLR.cc/2024/Conference — Submitted to ICLR 2024_

### Official Review · Reviewer_FurC · 2023-10-31

**Soundness:** 2 fair
**Presentation:** 2 fair
**Contribution:** 2 fair
**Rating:** 5
**Confidence:** 4

**Summary:**

This work introduces a gradient-based model for prompt optimization in text-to-image generation. Experiments are performed over collected prompts to evaluate the proposal.

**Strengths:**

++ Both adversarial attack and prompt improvement tasks are included for evaluation.

++ The main idea is novel and interesting.

**Weaknesses:**

-- The figure 1 should be professionally re-shaped by highlighting the main contributions, instead of listing each block.

-- Only one metric of CLIP loss is used for evaluation, which makes the experimental results somewhat unconvincing. As pointed in [A], clip score does not correlate well with human choices. It is necessary to report the results under more metrics (e.g., Aesthetic Score, Human Preference Score [A], or Human Preference Score v2 [B]).

Moreover, the prompt dataset in this work contains only 300 prompts for each task. It is better to evaluate the  3200 prompts on HPS benchmark [B].

[A] Human Preference Score: Better Aligning Text-to-Image Models with Human Preference, ICCV 2023.
[B] Human Preference Score v2: A Solid Benchmark for Evaluating Human Preferences of Text-to-Image Synthesis[J]. arXiv preprint arXiv:2306.09341, 2023.

-- Authors select an earlier version of Stable Diffusion 1-4 as the base model. I am curious to see the results when applying the proposal into superior version of Stable Diffusion (SDXL).

-- It is not safe to evaluate the prompt improvement task with clip loss in Table 1 (b), where the state-of-the-art method (Promptist) even shows worse performance than user input. I checked the Promptist paper, where extensive human study is performed to validate the effectiveness compared to user input. Thus I have to say the comparison in Table 1 (b) is somewhat not fair, and this work does not give enough credit to existing work of Promptist.

Moreover, following Promptist paper, it is also necessary to perform human study to evaluate the proposal.

-- Examples in Figure 2 seem to be cherry-picked.

**Questions:**

Please check the details in Weaknesses section

---

> ### Author Response · Authors · 2023-11-20
> **Response to R4 - FurC - Part 1/3**
>
> Thank you for your positive feedback. We hope the responses below could address your questions and concerns. To make the responses cleaner, we categorized your comments and answer them by topics. If you have any further questions, please feel free to let us know, and we are more than happy to discuss with you!
>
> **Concern on evaluation metrics:**
>
> *“Only one metric of CLIP loss is used for evaluation, which makes the experimental results somewhat unconvincing. As pointed in [A], clip score does not correlate well with human choices. It is necessary to report the results under more metrics (e.g., Aesthetic Score, Human Preference Score [A], or Human Preference Score v2 [B]).* It is not safe to evaluate the prompt improvement task with clip loss in Table 1 (b), where the state-of-the-art method (Promptist) even shows worse performance than user input. I checked the Promptist paper, where extensive human study is performed to validate the effectiveness compared to user input. Thus I have to say the comparison in Table 1 (b) is somewhat not fair, and this work does not give enough credit to existing work of Promptist. Moreover, following Promptist paper, it is also necessary to perform human study to evaluate the proposal.”
>
> **[Response]**
>
> Thank you for your suggestion. We agree that the lack of a powerful automatic evaluation method is a limitation faced by the text-to-image generation field as a whole, and there is currently no perfect metric, including CLIP score. This is also why our original paper included human evaluation results to compensate it.
>
> - Automated evaluation
> 1. **CLIP Score: While clip is not perfect, it is still the most widely-adopted automated metric to judge text and image similarity**, and hence for evaluating prompt following ability of diffusion models. For instance, it has been adopted in many existing works as the de facto metrics in this aspect, such as DALLE-3 [1], Promptist [2], BlendedDiffusion [3], etc. So we also followed the pioneers to adopt it as the automated metrics.
> 2. Aesthetic score: Aesthetic is an important aspect for evaluating image generation for artistic applications. However, **the main focus of our experiments is on the prompt following ability, and we do not optimize for aesthetics** like Promptist or MagicPrompt. In our paper, we respectfully acknowledged the effectiveness of Promptist in generating aesthetically pleasing images (Section 2 para 2). On this matter, we follow DALLE-3 to view prompt following and aesthetics as distinct aspects of image generation that can be evaluated and reported separately [1].
> 3. Human Preference Score v2: We included results on HPSv2 score in the Appendix F; Table 3 suggests that the performance of DPO-Diff is consistent on HPSv2. Note that HPSv2 entangles prompt-following ability and aesthetic ability into a single score; **Although we only optimize for prompt following ability, it is interesting to see that the proposed method also performs well when evaluated with HPSv2.**
> - Human evaluation
>
> **The human study on prompt optimization is already included in the original submission**, and discussed in Section 5.1, 5.3 and Appendix D.2. The practice of reporting quantitative results using an automatic metric and and human study is borrowed from previous works such as Promptist and DALLE-3. Moreover, following the suggestion of R3, **we also conducted the human evaluation on adversarial attack.** The result is summarized in the Appendix F, where DPO-Diff obtains a success rate of 44%.
>
> We hope that the above discussion could address your concern on the evaluation metrics.

---

> ### Author Response · Authors · 2023-11-20
> **Response to R4 - FurC - Part 2/3**
>
> **Concern on the evaluation dataset size**
>
> *“Moreover, the prompt dataset in this work contains only 300 prompts for each task. It is better to evaluate the 3200 prompts on HPS benchmark [B].”*
>
> **[Response]**
>
> Thank you for your suggestion.
>
> 1. We would like to clarify that, the 600 prompts used in our experiments are “hard prompts” filtered from **a pool of roughly 6000 prompts** (from coco, diffusiondb and chatgpt), as mentioned in Section 5.1. Concretely, for prompt improvement (adversarial attack) task, we focus on prompts that are hard (easy) for the diffusion model to follow. The filtering is conducted automatically by thresholding CLIP loss on the original Stable Diffusion model, without favoring a certain method.
> 2. Moreover, we found that **the statistics tend to stabilize after 100 prompts** (providing they are sampled from diverse sources). We conjecture that this is probably one reason why recent works typically only use a few hundred prompts, such as 170 for DALLE-3 and 400 for LLM-grounded Diffusion when evaluating the prompt following ability, as this is more resource and labor friendly (human evaluators).
>
> ---
>
> **Concern on Stable Diffusion version**
>
> *“Authors select an earlier version of Stable Diffusion 1-4 as the base model. I am curious to see the results when applying the proposal into superior version of Stable Diffusion (SDXL).”*
>
> **[Response]**
>
> Thank you for your suggestion. We originally picked v1-4 since this is most commonly used in academia due to resource efficiency. We also experimented with v2 and v1-5, but the results are very similar to v1. **We are currently repeating the prompt improvement experiments in SDXL.** As “hard prompts” for SDXL are different than v1-4, we need to rerun the data collection as well. Since SDXL is nearly 10x slower than v1 and can only be run on the very limited number of 48G GPUs we have, the final results will be delayed. We will try to update them in our revision if they can be finished before the end of the rebuttal. But based on the current progress, we observe the following preliminary results:
>
> 1. SDXL demonstrates improved prompt following ability than SD v1-4 or v2, as there are fewer hard prompts.
> 2. **The improvement of DPO-Diff over baselines seems consistent on SDXL.**

---

> ### Author Response · Authors · 2023-11-20
> **Response to R4 - FurC - Part 3/3**
>
> **Concern on comparison with Promptist**
>
> *”It is not safe to evaluate the prompt improvement task with clip loss in Table 1 (b), where the state-of-the-art method (Promptist) even shows worse performance than user input. I checked the Promptist paper, where extensive human study is performed to validate the effectiveness compared to user input. Thus I have to say the comparison in Table 1 (b) is somewhat not fair, and this work does not give enough credit to existing work of Promptist.”*
>
> **[Response]**
>
> Thank you for raising this point. We would like to first clarify how we perform comparisons with baselines, then share our view on the relationship between DPO-Diff and Promptist.
>
> - Comparison with baselines
>
> As mentioned in the first reply, **the original paper included the human evaluation results as well**.  Note that since the focus of our experiments is solely on the prompt following ability of diffusion models, we also prompt our human judgers to only evaluate the text and image similarity. Note that Promptist optimizes for a combined objective of aesthetic and text-image similarity (also used CLIP), so the evaluation in the original paper entangled them. Based on our experience with Promptist, we found that it can almost always generate aesthetically pleasing images, yet the prompt following ability is rather lacking [cite]. We hypothesize that this behavior is primarily caused by the lack of a paired dataset, rather than the RLHF method it uses, as RLHF has been shown great success in LLM. And in the limitation section, we point out that DPO-Diff can be used to help with it (Appendix A: Limitation).
>
> - Connection between DPO-Diff and Promptist
>
> **DPO-Diff can be used to generate offline paired dataset for Promptist.** As pointed out in our paper (Section 2), the performance Promptist is currently bottlenecked by the lack of high-quality paired prompt data to be used for RLHF. We believe this is the reason why Promptist excels at aesthetics but struggles with prompt following, as the dataset it trains on primarily consisting of (user prompt, context-agnostic aesthetic modifiers) pairs. And it is a known fact in LLM that RLHF requires high quality instruction data. On the other hand, the advantage of Promptist is its inference efficiency, as we acknowledged in the limitation section. Therefore, DPO-Diff can be used as an offline method to generate synthetic data for Promptist to train on. This way, the result combines the power of both worlds. In fact, we have been discussing this strategy internally as a natural followup project.
>
> **Therefore, We view DPO-Diff as a novel framework orthogonal to instruction-finetuning paradigm (Promptist)**: Each has their unique pros and cons and compensates each other very well. We compare with Promptist because it is the only prompt optimization method for text-to-image diffusion, even though DPO-Diff and Promptist are actually orthogonal. Despite the orthogonality, we still compare with Promptist, because it is the first and only existing prompt optimization method for text-to-image diffusion. Although prompt optimization for diffusion models is arguably more challenging, prompting LLM has been receiving much more attention than diffusion model. So we believe it reveals the urgency of promoting efforts in developing prompt optimization algorithms for diffusion models.
>
> **In conclusion, the DPO-Diff is designed not as a replacement for Promptist, but an orthogonal and complementing method.** Yet despite the orthogonality, we still compare with Promptist, because it is the first and only existing prompt optimization method for text-to-image diffusion. As pointed out in the introduction, efforts on prompt optimization for diffusion model is almost negligible compared with LLM. So we feel it reveals the urgency of promoting efforts in this area at the current stage, which will be beneficial to the community.
>
> ---
>
> **Suggestion 1: show changes in Figure 1**
>
> *“The figure 1 should be professionally re-shaped by highlighting the main contributions, instead of listing each block.”*
>
> Thank you for your suggestion! We’ve modified Figure 1 to show the inference process of obtaining text gradient with and without shortcut gradient.
>
> [1] Betker et al. Improving Image Generation with Better Captions. 2023
>
> [2] Hao et al. Optimizing Prompts for Text-to-Image Generation. 2022
>
> [3] Avrahami et al. Blended Diffusion for Text-driven Editing of Natural Images. CVPR 2022

---

> ### Author Response · Authors · 2023-11-23
> **Response - SDXL exp finished, with consistent results as those on v1-4**
>
> Dear R4 - FurC
>
> We've updated the results on SDXL in the revision, including quantitative evaluation under CLIP and HPSv2, human evaluation, and sample images. While SDXL is overall a step ahead of SD v1-4 in image quality, it still struggles with compositional generation like earlier versions, as it still misses objects and falsely binds attributes with objects. We found that the results across various evaluations are consistent with what DPO-Diff achieved on v1-4.
>
> This would conclude all requested experiments from your initial review. We hope the response could improve your view of the paper. And we are eager to hear back from you.
>
> DPO Authors.

---

### Official Review · Reviewer_emXz · 2023-10-31

**Soundness:** 3 good
**Presentation:** 3 good
**Contribution:** 2 fair
**Rating:** 5
**Confidence:** 4

**Summary:**

This paper explores the domain of text-to-image diffusion models with a focus on a gradient-based prompt optimization framework. Given the vast optimization spaces in text and the memory inefficiency of computing text gradients, the author introduces two innovative workarounds to address these challenges. First, they create a compact subspace that is most relevant to the user input. Then, they propose the use of "Shortcut Gradient," which abstains from gradient computation for large timesteps and instead conducts smaller steps. Finally, the paper suggests predicting the final image through denoising from the last gradient-computed timestep. This approach is versatile and can be employed for tasks such as generating adversarial prompts for model diagnosis or enhancing prompts to improve user-input fidelity.

**Strengths:**

A noteworthy strength of this paper is its investigation into the impact of prompts on text-to-image diffusion models. It contributes to the field by improving prompt optimization efficiency and introduces several techniques to address this issue.

**Weaknesses:**

While the proposed method effectively tackles the challenges of prompt optimization, it does rely on several heuristics, which could be considered a limitation. For instance, the "Shortcut Gradient" method necessitates tracking the CLIP loss throughout timesteps, which introduces complexity. Moreover, the negative prompt library relies on human-crafted prompts, potentially making it highly dependent on the choice of diffusion models. Another limitation is evident in Table 1, where the evaluation solely reports the CLIP loss. The absence of metrics measuring the relevance between the modified prompt and the user-provided prompt raises concerns about the comprehensiveness of the evaluation. Additionally, in adversarial prompt experiments, the evaluation is limited to the CLIP loss, which may not capture the full scope of the method's performance, considering its primary focus on maximizing CLIP loss.

**Questions:**

The paper raises questions about comparisons with the "Hard Prompts Made Easy" paper [https://arxiv.org/abs/2302.03668]. It would be valuable to understand how the proposed DPO-Diff method performs in comparison to simply optimizing the CLIP loss. This comparison could shed light on the relative advantages and disadvantages of each approach.

In Table 1, it would be helpful if the author could provide more context regarding the difference in the number of User Inputs for Table 1 (a) and Table 1 (b). An explanation of the reasons behind this discrepancy in the experimental setup would enhance the reader's understanding of the results.

---

> ### Author Response · Authors · 2023-11-20
> **Response to R3 - emXz - Part 1/2**
>
> Thank you for your positive feedback. We hope the responses below could address your questions and concerns. If you have any further questions, please feel free to let us know, and we are more than happy to discuss with you!
>
> **Concern 1: rely on tracking CLIP loss throughout timesteps**
>
> *“For instance, the "Shortcut Gradient" method necessitates tracking the CLIP loss throughout timesteps, which introduces complexity.”*
>
> **[Response]**
>
> Thank you for the comment. We would like to provide some clarification on the CLIP loss part of the Shortcut Gradient:
>
> 1. Putting efficiency aside, applying a loss function on the generated image and backpropagating it through the diffusion inference steps is ***not* fundamentally different from backpropagation in any other network**. As mentioned in the related work, for diffusion models, this has also demonstrated positive results on multiple applications before, such as sampler learning and adversarial purification [1, 2]. In this sense, adopting CLIP loss in DPO-Diff does not introduce any extra complexity.
> 2. The main obstacle is how to make the backpropagation memory and runtime efficient as a naive solution requires $50x$ backward pass of the rather large UNet in Stable Diffusion. Prior efforts [1, 2] achieve constant memory at the cost of even longer runtime, therefore limiting their application to only toy Diffusion models.
> 3. **The main technical contribution of DPO-Diff in this particular area is a generic computation method (Shortcut Gradient) to reduce this complexity to constant memory and runtime for the first time, with impact on many applications beyond prompt optimization (e.g. the applications considered in previous works on toy models).**
> 4. Further, the closed-form estimation of $x_0$ from $x_t$ used in Shortcut Gradient is a principled solution that can be derived from DDPM (Remark 1 in our paper, with its derivation added to the Appendix).
>
> We hope that the above clarification could address your concern; but if we misunderstand your initial point in any way, please let us know.
>
> ---
>
> **Concern 2: rely on human-crafted negative prompts**
>
> *“the negative prompt library relies on human-crafted prompts, potentially making it highly dependent on the choice of diffusion models”*
>
> **[Response]**
>
> Thank you for the detailed review. **DPO-Diff actually works equally well when solely relying on the Antonym Space**, that is, the Negative Prompt Library can be removed with negligible effect on the performance. The main reason we included those human-crafted negative prompts is purely for practical purposes:
>
> 1). It allows DPO-Diff to be used (to some extent) offline when the user does not have access to OpenAI’s API (it’s a paid service).
>
> 2). It resolves the situation where the user prompt has only 1-2 words. This occasionally occurs on DiffusionDB, since its web-crawled prompts are highly noisy.
>
> ---
>
> **Concern 3:**
>
> *“The absence of metrics measuring the relevance between the modified prompt and the user-provided prompt”*
>
> **[Response]**
>
> Thank you for your suggestion.
>
> 1. For prompt improvement tasks, since we only search for negative prompts, the user prompt is unmodified. (You might be aware of this already, but we will still put it here for the completeness of the response)
> 2. For adversarial attack tasks, we did not report the user and adversarial prompts’ similarity because we already **adopted a thresholding scheme**: During the search, we threshold the cosine similarity between user and adversarial prompts in T5 embedding space, to eliminate false positive adversarial prompts caused by wrong synonyms generated occasionally by ChatGPT. The threshold was set to 0.9 for all datasets. We’ve added a discussion on this in the Appendix C.2. Moreover, when conducting human evaluation, we also asked the human evaluator to further identify false-positive adversarial prompts that changes the meaning of user prompt (Appendix F).

---

> ### Author Response · Authors · 2023-11-20
> **Response to R3 - emXz - Part 2/2**
>
> ---
>
> **Concern 4:**
>
> *“in adversarial prompt experiments, the evaluation is limited to the CLIP loss, which may not capture the full scope of the method's performance, considering its primary focus on maximizing CLIP loss.“*
>
> **[Response]**
>
> Thank you for your suggestion. We should’ve also included human evaluation on this task for future reference —- we added this results in the Appendix F. Since there is no comparable baseline in this task, we only compare with the user prompt baseline. The result shows that DPO-Diff obtained a success rate of 44% according to human evaluation. Note that this rate excludes the case where a success is achieved only because the adversarial prompt changes the meaning of user prompt.
>
> ---
>
> **Question 1: Comparison with PEZ**
>
> *“The paper raises questions about comparisons with the "Hard Prompts Made Easy" paper [[https://arxiv.org/abs/2302.03668]](https://arxiv.org/abs/2302.03668%5D). It would be valuable to understand how the proposed DPO-Diff method performs in comparison to simply optimizing the CLIP loss. This comparison could shed light on the relative advantages and disadvantages of each approach.”*
>
> **[Response]**
>
> Thank you for your suggestion and the pointer. This seems an interesting work. We wanted to include this method in the evaluation, but we ran into issues running the provided code. Since the duration for the rebuttal period is limited, we will try to incorporate this method in the future revision. Based on our understanding, for the application of diffusion model, PEZ learns the optimized prompt directly on the clip VLM encoder. This is achievable because Stable Diffusion also uses pretrained CLIP text encoder. Intuitively, we think that one advantage of PEZ could be computational speed, since backpropagation only goes through the text encoder. The downside is that PEZ’s optimization objective is not directly related to stable diffusion as a whole, as it ignores image generation pathway entirely. Moreover, if we understand correctly, PEZ might not apply to diffusion models whose encoder is trained purely on language tasks, i.e. without a visual pathway (e.g. Imagen with T5 encoder).
>
> ---
>
> **Question 2: Different clip scores of user input between Table 1a and tb**
>
> *“In Table 1, it would be helpful if the author could provide more context regarding the difference in the number of User Inputs for Table 1 (a) and Table 1 (b). An explanation of the reasons behind this discrepancy in the experimental setup would enhance the reader's understanding of the results.”*
>
> Thank you for your suggestion. By “the number of User Inputs for Table 1(a) and 1(b)”, we assume that you are referring to the discrepancy in clip loss? In this case, the difference is caused by “hard prompt filtering” explained in the experimental setup section (5.1 Dataset preparation). Concretely, the definition of hard prompts for adversarial attack and prompt improvement are reversed: in adversarial attack, we want to attack the good prompts (low clip loss), whereas in prompt improvement, we want to improve the poor prompts (high clip loss). But we will add it to the Table caption as well for ease of reference.
>
> [1] Watson et al. Learning Fast Samplers for Diffusion Models by Differentiating Through Sample Quality. ICLR 2022
>
> [2] Nie et al. Diffusion Models for Adversarial Purification. ICML 2022.

---

> > ### Comment · Reviewer_emXz · 2023-11-23
> > **Response**
> >
> > Thank you for the authors for rebuttal.
> >
> > My concerns are mostly addressed after the rebuttal, and I believe the proposed method has technical novelty. However, I still believe there is a room for improvement in evaluation and conveying the effectiveness of the method. Therefore, I will keep my rating.

---

### Official Review · Reviewer_3K8N · 2023-11-01

**Soundness:** 3 good
**Presentation:** 3 good
**Contribution:** 2 fair
**Rating:** 5
**Confidence:** 3

**Summary:**

This paper presents a systematic study of prompt optimization for text-to-image diffusion models. Two tricks are proposed to solve two challenges: 1) searching for synonyms/antonyms to mitigate the problem of enormous domain space; 2) shortcut gradient to mitigate the problem of backpropagating the inference chain. The proposed methods are straightforward but results are promising.

**Strengths:**

- The paper studies an important problem of prompt optimization. The paper, to the best of the authors' knowledge, provides the first exploratory work on automated negative prompt optimization.
- The presented results look promising.

**Weaknesses:**

- Clarification question: how are the compact domain space contacted and shortcut gradient related? Do we dynamically select words to search for synonyms/antonyms based on the gradients?
- Clarity: 1) The asterisk symbol in eq 6 seems not usually used in this case; 2) it seems that the main issue with backproping through the entire inference chain is the prohibitive memory cost, so abbreviating it as "text gradient" is a little misleading and inaccurate.
- Shortcut gradient: 1) is it necessary to use estimated x_0 from t-K? is there any ablation to support the claim? 2) would strategies like randomizing the truncation lengths [1] be helpful in this case?

[1] Prabhudesai, Mihir, et al. "Aligning Text-to-Image Diffusion Models with Reward Backpropagation." arXiv preprint arXiv:2310.03739 (2023).

**Questions:**

How can we make sure that the synonym-swapped sentence is still semantically similar to the original sentence and how do we choose the distance d? e.g. in Fig 2 (a) "grease picture" is different from "oil painting".

---

> ### Author Response · Authors · 2023-11-20
> **Response to R2 - 3K8N - Part 1/2**
>
> Thank you for your positive feedback. We hope the responses below could address your questions and concerns. If you have any further questions, please feel free to let us know, and we are more than happy to discuss with you!
>
> **Concern 1: clarify the relationship between compact domain space and shortcut gradient**
>
> *Clarification question: how are the compact domain space contacted and shortcut gradient related? Do we dynamically select words to search for synonyms/antonyms based on the gradients?*
>
> **[Response]**
>
> Thank you for the question. We feel that briefly summarizing the pipeline might answer your questions. The DPO-Diff framework is formulated as an optimization problem (eq.4), which includes a domain (a.k.a. search space) that includes potential candidates, and an objective that need to be solved over the domain space. The entire pipeline can be summarized in chronological order as:
>
> - Domain Space: We define the domain space as the compact Synonym/Antonym Space generated for each user prompt. So these compact spaces are dynamic only in the sense that they differs for each user prompt. Once the domain space is generated for a given user prompt, it will be kept fixed through the optimization process.
> - Given the domain space generated above, we use the proposed gradient-based algorithm (GPO) to search for a good prompt from the domain space. The gradient used in this algorithm is obtained via “Shortcut Gradient”.
>
> To further improve the clarity of the paper, we’ve added the above summarization pipeline to beginning of Section 4.
>
> ---
>
> **Concern 2 on Clarity of the equation and “text gradient”**
>
> “*Clarity: 1) The asterisk symbol in eq 6 seems not usually used in this case; 2) it seems that the main issue with backproping through the entire inference chain is the prohibitive memory cost, so abbreviating it as "text gradient" is a little misleading and inaccurate.”*
>
> **[Response]**
>
> 1. Asterisk
>
> Thank you for pointing it out. Yes, since this is just a multiplication sign, it can indeed be omitted here. We’ve modified the equation.
>
> 1. Using the term “Text Gradient”
>
> You are right that the main issue with backpropagating through the inference process is the memory and runtime cost. We use the term “text gradient” because we only deal with the gradient over discrete text prompts to solve the optimization problem. We then use the term “shortcut gradient” to refer to our method to efficiently obtain the “text gradient”. In terms of “shortcut gradient”, **it is indeed a much more general method than just computing “text gradient”, as it can be used to compute gradients w.r.t. any model components (weights, embedding, any conditional signal) through the inference steps.** We hope that could address your concern, but please let us know in case we misinterpreted your point.

---

> ### Author Response · Authors · 2023-11-20
> **Response to R2 - 3K8N - Part 2/2**
>
> ---
>
> **Concern 3: necessity of estimating $x_0$ from $x_t$**
>
> “*Shortcut gradient: 1) is it necessary to use estimated x_0 from t-K? is there any ablation to support the claim? 2) would strategies like randomizing the truncation lengths [1] be helpful in this case?”*
>
> **[Response]**
>
> Thank you for raising this important point. Yes, it is necessary to estimate $x_0$ from $x_t$. The justification is as follows. Suppose we are currently located at a randomly sampled step $t$ between $T \sim 0$:
>
> 1. If we do not estimate $x_0$ from $x_t$, we will have to run the UNET of the diffusion model $t$ times to obtain the final $x_0$ and then propagate the loss on $x_0$ all the way back from step $0$ to step $t$. This is exactly what Randomlized Truncated Backpropagation (RTB) did, and the memory and runtime complexity becomes $\mathcal{O}(t)$.
> 2. If $t$ is uniformly sampled from $T \sim 0$, the complexity becomes $\mathcal{O}(t) = \mathcal{O}(T/2) = \mathcal{O}(T)$, which is still the same as brute-forcely backpropagate.
> 3. For RTB, the only way to reduce its theoretically linear complexity in practice is to pick a constant $t$ near the final image $x_0$. This means to truncate most but the last few inference steps. On GPUs with 48G memory, we found that the maximum $t$ it can support is only $t=3$ (out of 50 steps) for Stable Diffusion v1-4 (let alone larger ones such as SDXL). However, the gradient signal on text is not informative for such a small $t$, since the image is almost finalized at this point. To show this, we added an ablation on the gradient norm across timesteps in Figure 6 in the Appendix, where it shows that gradient norm near $t=0$ is almost negligible compared with the middle timesteps.
>
> Hence, we conclude that estimating $x_0$ from $x_t$, as done in Shortcut Gradient, is critical in obtaining an informative text gradient efficiently. It is the only current solution to this problem, and for the first time makes gradient-based optimization on the inference steps of large-scale diffsion models practical.
>
> Furthermore, we would like to make the following remarks to the above response:
>
> 1). **RTB and Shortcut gradient solve different problems.** RTB is proposed to solve the over-optimization problem [cite their paper], and it still incurs linear runtime and memory complexity, while Shortcut Gradient is proposed for efficiency.
>
> 2). **Our closed-form estimation of $x_0$ from $x_t$ is a principled solution grounded in the derivation of DDPM** (Remark 1 in the paper, with derivation added to the Appendix).

---

### Official Review · Reviewer_BFJd · 2023-11-03

**Soundness:** 3 good
**Presentation:** 2 fair
**Contribution:** 2 fair
**Rating:** 5
**Confidence:** 3

**Summary:**

This paper introduces the first gradient-based framework for optimizing prompts for text-to-image diffusion models. Two main challenges are addressed: 1) the enormous search space of possible prompts, and 2) the high computational cost of computing text gradients through the full diffusion model. To tackle these issues, dynamically generated compact subspaces of only the most relevant words are used to restrict the search space. A "Shortcut Gradient" is introduced to efficiently approximate the true text gradient with constant memory and runtime. Experiments show the framework can enhance or adversarially attack diffusion model faithfulness by discovering improved or destructive prompts respectively. Overall, this represents a novel gradient-based approach for prompt optimization that restricts the search space and enables efficient approximation of text gradients. The key innovations are the compact subspaces and Shortcut Gradient which make prompt optimization tractable.

**Strengths:**

1. The paper identifies an important problem that is not well-addressed
2. The paper clearly shows two challenges and proposes effective solutions and the results look promising
3. The experiments seem solid to support the claims on multiple sources and tasks.

**Weaknesses:**

1. The paper is not well written and easy to follow. Some core sections are not fully explained.
2. The paper structure in the method section can be better organized to clearly show the components step by step
3. The paper provides several remarks, and definitions but does not present more proof with detailed discussions.
4. All of the above issues make this work hard to be reproduced.
5. The experiments lack important baselines for comparison, such as instructzero [1]
6. The ablation study can be further improved with more discussions.

[1] Chen, Lichang, Jiuhai Chen, Tom Goldstein, Heng Huang, and Tianyi Zhou. "InstructZero: Efficient Instruction Optimization for Black-Box Large Language Models." arXiv preprint arXiv:2306.03082 (2023).

**Questions:**

Most of questions are listed in the weakness section

---

> ### Author Response · Authors · 2023-11-20
> **Response to R1 - BFJd**
>
> Thank you for your positive feedback. We hope the responses below could address your questions and concerns. If you have any further questions, please feel free to let us know, and we are more than happy to discuss with you!
>
> **Concern 1,2,3,4,6 - Clarity and code release**
>
> “*The paper is not well written and easy to follow. Some core sections are not fully explained. The paper structure in the method section can be better organized to clearly show the components step by step. The paper provides several remarks, and definitions but does not present more proof with detailed discussions. All of the above issues make this work hard to reproduce. The ablation study can be further improved with more discussions.”*
>
> **[Response]**
>
> - Reproducibility
>
> We're planning to release all our code, since we view our method only as a starting point for future advancement. **For the rebuttal, we attached a draft version of the codebase to the supplementary material.** We are actively working on refactoring the codebase and are also looking into adding a demo on HuggingFace.
>
> - Clarity
>
> Thank you for your detailed suggestions! We’ve improved the presentation accordingly. The roadmap of the current manuscript is as follows: we first explain the framework in Section 4.1. This is followed by the search space design in Section 4.2 and the search algorithm for navigating that search space in Section 4.3. Within Section 4.3, we roughly follow the order of execution to lay out the complete algorithm, that is, we discuss how to learn a distribution over text space (GPO) first, then how to sample from it (evolutionary search).
>
> **To improve the clarity of the paper, we made the following changes based on your suggestion:**
>
> 1. \+ The above roadmap to the beginning of Section 4.
> 2. \+ Discussions on Definition 4.1 in the Appendix
> 3. \+ Derivation for Remarks 1 and 2 in the Appendix.
> 4. \+ Extended discussion on the ablation study results in the Appendix.
>
> We also welcome any further suggestions you may have, as we are also eager to make this rather dense work more reader friendly.
>
> ---
> **Concern 5 - Lack of important baselines**.
>
> “*The experiments lack important baselines for comparison, such as instructzero.”*
>
> **[Response]**
>
> Your concern actually points out a critical unbalance of this field - there is much more effort in developing prompt optimization for LLMs compared with diffusion models. To the best of our knowledge, Promptist is the first and the only comparable method developed for and tested on the text-to-image diffusion model. This is also the reason why in our opinion, efforts such as Promptist and ours for the diffusion models might be valuable to the community.
>
> Prompt optimization methods like APE [1] and InstructZero [2] are designed for LLM, and their common formulation makes them not directly extendable to diffusion models. Concretely, they mainly study discriminative tasks and assumes a labeled training dataset (demos) is provided [1, 2]. This is one of the reasons that these methods do not report experiments on diffusion models.
>
>
> [1] Zhou et al. Large Language Models Are Human-Level Prompt Engineers. ICLR 2023
>
> [2] Chen et al. InstructZero: Efficient Instruction Optimization for Black-Box Large Language Models. 2023

---

### Author Response · Authors · 2023-11-20
**Common reply to all reviewers**

Firstly, we would like to express our gratitude to all of our reviewers for their efforts in reviewing our paper. We are particularly encouraged by the positive feedbacks we received on the importance of the problem addressed (R1, R2, R3), originality (R1, R2, R3, R4) and versatility (R3, R4) of the framework, as well as empirical results (R1, R2)

**Our reviewers also raised several concerns in the weakness section, which in fact are primarily providing valuable suggestions and asking for clarification**; Both of them would greatly help us in refining the paper and strengthening the results. We made an effort to address our reviewers’ questions in the following individual responses, and also incorporated the discussions and suggested experiments in the revision.

To avoid messing up the existing references during the rebuttal period, we append most of the new contents to the appendix (marked blue), but will move them to the main text later. Below are the **highlighted changes to the revision:**

1. [CODE] + Released the draft code for reference, we will post it on github and huggingface after finished with cleaning. (R1)
2. [TEXT] + a discussion on definitions and derivations of remarks. (R1)
3. [TEXT] + Extended the discussion on ablation study in the Appendix (R1)
4. [EXP] + an ablation showing the gradient norm v.s. timesteps. (R2)
5. [EXP] + HPSv2 score as an automated metric, in addition to CLIP score. (R4)
6. [EXP] + experiments on the StableDiffusion-XL, the state-of-the-art open-sourced T2I diffusion model. (R4)
7. [EXP] + human evaluation on adversarial attack. (R3)

We hope our response can help you in finalizing the ratings of our paper. If you have any other questions, please feel free to reply back, and we will answer them asap!

Sincerely,

DPO-Diff Author

---

### Author Response · Authors · 2023-11-22
**Look forward to your response**

Dear Reviewers,

Since there is only one day left for the reviewer-author discussion phase, we want to remind you that we are eager to hear from your feedback about our rebuttal and catch any further questions. Thank you!

Best regards,
Authors

---

### Author Response · Authors · 2023-11-23
**SDXL results included, and improvements of DPO-Diff on this SOTA variant is consistent**

Dear Reviewers,

This is a gentle notification that we've included the experiments on SDXL as suggested by R4 - FurC. The results are included in the Appendix. While SDXL is overall a step ahead of SD v1-4 in image quality, it still struggles with compositional generation like earlier versions, as it still misses objects and falsely binds attributes with objects. We found that the results across various evaluations (CLIP score, HPSv2, Human Evaluation, Qualitative results) are consistent with what DPO-Diff achieved on v1-4.

We value your suggestions and have incorporated them in the revision; we hope that the revision and response will improve your view of the paper.

Best,
DPO Authors.

---

### Meta-Review · Area_Chair_hDpS · 2023-12-05

**Metareview:**

The paper proposes a gradient-based framework for prompt optimization in text-to-image diffusion models. It is formulated as a discrete optimization problem and shows improvements under CLIP loss.

Initially, the major concern of the reviewers was about the evaluation, i.e. the CLIP score. The CLIP loss was optimized but it can be misleading.

The authors provided rebuttal and I encouraged the reviewers to discuss with the authors and each other. Unfortunately, only reviewer 3K8N replied to the authors and kept the score.

Since all reviewers voted a score of 5 and the original submission missed important justifications, I tend to reject this paper. Nevertheless, I think this is a promising paper if the authors improve it according to the comments of the reviewers.

**Justification For Why Not Higher Score:**

Since all reviewers voted a score of 5 and the original submission missed important justifications, I tend to reject this paper.

**Justification For Why Not Lower Score:**

N/A

---

### Decision · Program_Chairs · 2024-01-16

Reject